# Size limits the sensitivity of kinetic schemes

Jeremy A. Owen [1,5] ✉ & Jordan M. Horowitz [2,3,4] ✉

Living things benefit from exquisite molecular sensitivity in many of their key processes, including DNA replication, transcription and translation, chemical sensing, and morphogenesis. At thermodynamic equilibrium, the basic biophysical mechanism for sensitivity is cooperative binding, for which it can be shown that the Hill coefficient, a sensitivity measure, cannot exceed the number of binding sites. Generalizing this fact, we find that for any kinetic scheme, at or away from thermodynamic equilibrium, a very simple structural quantity, the size of the support of a perturbation, always limits the effective Hill coefficient. We show how this bound sheds light on and unifies diverse sensitivity mechanisms, including kinetic proofreading and a nonequilibrium Monod-Wyman-Changeux (MWC) model proposed for the *E. coli* flagellar motor switch, representing in each case a simple, precise bridge between experimental observations and the models we write down. In pursuit of mechanisms that saturate the support bound, we find a nonequilibrium binding mechanism, nested hysteresis, with sensitivity exponential in the number of binding sites, with implications for our understanding of models of gene regulation and the function of biomolecular condensates.

Sensitivity—the size of the response to a small perturbation—is a key figure of merit for performance on a number of tasks accomplished by living cells, including sensing chemical concentrations[1,2], accurate signal transduction in cascades[3], molecular discrimination[4–6], and gene regulation[7,8]. It is also a basic experimental observable, and so there is a long history of theoretical work connecting sensitivity measures to underlying mechanisms that could explain them—going back to Hill's realization that the sigmoidal binding curve of oxygen to hemoglobin[9] could be explained by binding to "aggregations" of hemoglobin[10].

The set of known mechanisms that can underlie high sensitivity is very diverse, growing to include—in the past 50 years—nonequilibrium ones such as the "futile cycle" and kinetic proofreading[4,11], whose study raises significant new challenges. Nevertheless, the success of a remarkably homogeneous modeling approach, rooted in chemical kinetics, makes possible a search for unifying principles—laws of sensitivity.

The prototypical sensitivity mechanism in biophysics is the cooperative binding of multiple copies of a ligand to a macromolecule.

The probability or fraction of the fully bound state is frequently fit with a Hill function[12],

$$f(x) = \frac{x^H}{K^H + x^H},\qquad(1)$$

where $x$ is the concentration of the ligand, $K$ is an effective dissociation constant, and the Hill coefficient, $H$ quantifies the (logarithmic) sensitivity. Equation (1) arises as an effective description in many different contexts, with $H$ depending in a complicated way on underlying details. However, in all cases of binding at thermodynamic equilibrium, there is a simple upper bound: the Hill coefficient cannot exceed the maximum number $n$ of ligands that can be bound at once. This limit on the sensitivity in terms of $n$ is purely structural, being independent of all affinities and kinetic parameters.

The bound on the Hill coefficient is just one example of the many tight links between structure and function that hold at thermodynamic equilibrium. But many challenges at the frontier of molecular biology today unavoidably require tackling nonequilibrium. For example, many aspects of gene regulation in eukaryotes[13–15]—from the spreading

[1]Department of Physics, Massachusetts Institute of Technology, Cambridge, MA 02139, USA. [2]Department of Biophysics, University of Michigan, Ann Arbor, MI 48109, USA. [3]Center for the Study of Complex Systems, University of Michigan, Ann Arbor, MI 48104, USA. [4]Department of Physics, University of Michigan, Ann Arbor, MI 48109, USA. [5]Present address: Department of Chemistry, Princeton University, Princeton, NJ 08540, USA. ✉e-mail: jo6038@princeton.edu; jmhorow@umich.edu

of epigenetic marks[16] to the action of enhancers[17,18]—have inspired the use of nonequilibrium models. These models confront us with many parameters we cannot measure or handle analytically. New, nonequilibrium laws relating structure to function would help us tackle this complexity.

In this work, we show how the equilibrium bound on the Hill coefficient admits a vast generalization to nonequilibrium systems. We find that for any kinetic scheme, the logarithmic sensitivity of any steady-state observable to a perturbation—as quantified, for example, by a Hill coefficient—cannot exceed the size of the support of the perturbation, a simple structural quantity we introduce: the support is the set of states that the system leaves faster after the perturbation than before. The size of the support is always less than the number of system states—the size of the kinetic scheme.

The support bound on sensitivity applies to a large class of models —all continuous-time Markov chains, sometimes known as "kinetic schemes" or "kinetic networks"—that are ubiquitous in biophysics, arising as the master equation of chemical reaction networks or as a coarse-grained description of the conformational dynamics of a single macromolecule. To illustrate the range of biological contexts in which the support bound applies, we show how it advances our understanding of a nonequilibrium Monod–Wyman–Changeux (MWC)-like model proposed for the *Escherichia coli* flagellar motor[19,20], recovers known limits to molecular discrimination in kinetic proofreading[5,6], and yields bounds on the accuracy of nonequilibrium chemical sensing[21,22]. In each of these examples, the support bound provides a way to go from experimental measurements of sensitivity to a concrete prediction about the underlying mechanism.

Finally, we apply the support bound to a class of models describing unordered, nonequilibrium, cooperative binding of a ligand (such as a transcription factor)—studied by prior authors[7,8,23] in the context of the highly sensitive Hunchback–Bicoid system[24] in *Drosophila*. The support bound yields an upper bound on the Hill coefficient exponential in the number of binding sites ($n$), exceeding the limits identified by numerical search of the parameter space[7,8]. We find that the exponential bound can in fact be achieved, by a nonequilibrium mechanism we identify and call nested hysteresis. The exponential-in-$n$ sensitivity achievable with nested hysteresis qualitatively exceeds that of any equilibrium sensitivity mechanism, with implications for the ascription of function to large molecular aggregations, such as biomolecular condensates, in the nonequilibrium context of a living cell.

## Results

### Kinetic schemes and perturbations

The sensitivity law we prove in this work applies to any system that may be modeled as undergoing transitions between a finite number of possible states $\{1, \ldots, N\}$, with transition rates depending (directly) only on the current state, not on history. Models of this form are ubiquitous in nonequilibrium physics, chemistry, and biophysics, where they are known by many names including: continuous-time Markov chains, Markov jump processes, kinetic networks[25], discrete-state kinetics[26], linear framework graphs[13,27], or as we will call them, kinetic schemes[28].

In any such model, the probability $p_i(t)$ for a system to be found in state $i$ at time $t$ evolves according to the master equation:

$$\frac{dp_i(t)}{dt} = \sum_{j=1}^{N} W_{ij} p_j(t), \qquad (2)$$

where $W_{ij}$ is the rate of the transition from state $j$ to state $i$, and the diagonal entries are $W_{jj} = -\sum_{i=1}^{N} W_{ij}$.

To any transition rate matrix $W$ can be associated a weighted, directed graph $G$ whose vertices are the states of the system and whose directed edges represent allowed transitions, weighted by the

transition rate. The structure of this graph $G$ plays a central role in the study of the scheme. In our figures (e.g., Fig. 1), we will liberally use drawings of the graph $G$ to represent schemes.

Under weak assumptions, the solution $p_i(t)$ to (2) converges to a unique steady-state distribution $\pi$ satisfying

$$\sum_{j} W_{ij} \pi_j = 0, \qquad \sum_{i} \pi_i = 1. \qquad (3)$$

Quantities measured experimentally are often averages over observation times long enough that transients can be neglected, so that what is really measured is just the average of some observable $A$ (any function of system state) over the steady-state distribution:

$$\langle A \rangle_\pi = \sum_i A_i \pi_i. \qquad (4)$$

Our focus in this work is how such steady-state averages, or ratios of them, respond to changes in a parameter of interest $x$ that controls some of the transition rates $W_{ij}(x)$—i.e., quantities of interest will always be of the form $f(x) = \langle A \rangle_\pi$ or $f(x) = \langle A \rangle_\pi / \langle B \rangle_\pi$. We will consider only positive observables, that is, observables $A$ for which each of the $A_i$ is nonnegative and at least one is positive.

We will also restrict attention to the case where the parameter $x$ simply multiplicatively scales some of the transition rates. This is often appropriate in biophysical examples for schemes representing the binding of (potentially several copies of) a ligand $L$ to a macromolecule, where $x$ is the concentration of $L$ (e.g., Fig. 1b). In this case, some transitions in a kinetic scheme will correspond to the binding of $L$, and the law of mass action leads to a linear, multiplicative dependence of the rate of those transitions on $x$.

The measure of sensitivity we focus on—and seek to bound—is the logarithmic sensitivity of a quantity of interest $f(x)$ with respect to the parameter $x$:

$$\frac{d \log f(x)}{d \log x} = \frac{x}{f(x)} \frac{df(x)}{dx}. \qquad (5)$$

What is the relationship between the derivative (5) and the Hill coefficient? If $f(x)$ were a Hill function (1), the logarithmic sensitivity would be

$$\frac{d \log f(x)}{d \log x} = H\left(\frac{K^H}{K^H + x^H}\right) = H(1 - f(x)), \qquad (6)$$

but there is no guarantee that a function of interest will actually "be" a Hill function. It is common nevertheless to report an "effective Hill coefficient," $H_{\text{eff}}$, but there are in fact several distinct quantities going by this name[8,12,19,29,30]. The different definitions all give $H_{\text{eff}} = H$ in the case of the Hill function, but in general they are not equivalent. However, we will see that a bound on the logarithmic sensitivity will also bound $H_{\text{eff}}$, in all its various incarnations. Note that a bound on the logarithmic sensitivity also implies a bound on the amplification of a "fold-change," another common sensitivity measure (Methods).

### The support bound

The support of the perturbation of $x$ is defined as the set of states (vertices in $G$) that have at least one outgoing transition (directed edge) that depends on $x$. The support consists of exactly those states whose exit rates depend on (and are increasing in) $x$. Note that the size of the support cannot exceed the total number of states $N$ in the scheme.

We are now ready to state our main result. If $A$ and $B$ are positive observables of a kinetic scheme, and $m$ is the size of the support of the

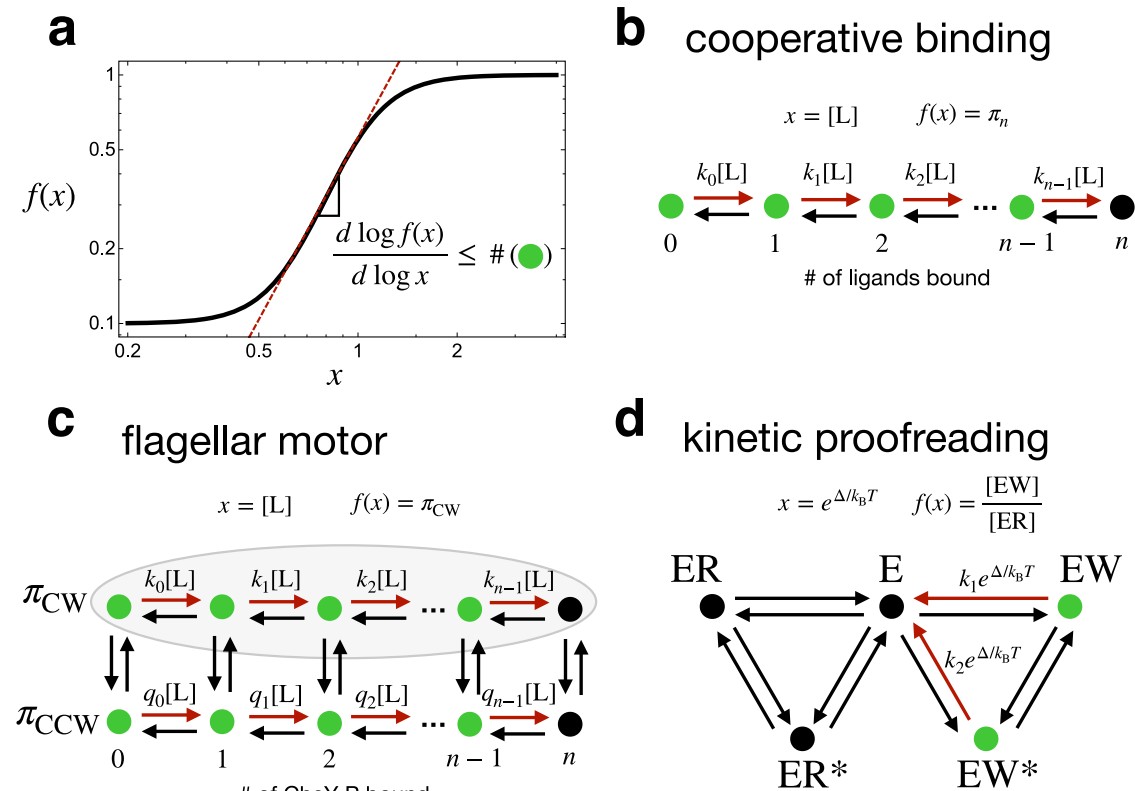

**Fig. 1 | Illustration of our main result. a** Example of a sensitive relationship between a parameter $x$ and a quantity $f(x)$. The slope on a log-log plot is a measure of sensitivity closely linked to the Hill coefficient. Our main result (inset equation) is that under general conditions this derivative is bounded by the size of the support of the perturbation of $x$. In each example (**b**, **c**, or **d**), the graph $G$ of a kinetic scheme, to which our result applies, is shown. The transitions whose rates depend on $x$ are indicated in red. The support (green) consists of those states from which the red transitions leave.

perturbation of $x$, then

$$\left| \frac{d \log \langle A \rangle_\pi / \langle B \rangle_\pi}{d \log x} \right| \le m. \tag{7}$$

Note that $m$ is completely independent of $A$ and $B$.

We call this inequality the support bound. Our proof, which we give in the Methods, is an application of the Markov chain tree theorem[31–36], which gives, for any kinetic scheme, an explicit algebraic expression for the steady state $\pi$ in terms of all the transition rates. Our result (7) refines prior results in the Markov chain literature[37,38] which gave bounds on sensitivities in terms of the total number of states $N$. It is also related in spirit to the results of Wong et al.[39], who apply the Markov chain tree theorem to find structural conditions for the emergence of the Michaelis–Menten formula from general kinetic schemes.

The inequality (7) serves as a companion result to those of our own prior work[40], which aimed to understand nonequilibrium response subject to thermodynamic constraints. The support bound reveals the limits of sensitivity set by structure alone, when thermodynamic constraints are completely loosened.

A useful corollary of (7) follows from taking the observable $A$ to be the indicator function of a subset $X$ of states and $B$ to be the indicator function of the complement $\overline{X}$ of $X$. In this case, $\langle A \rangle = \pi_X$ is the steady-state probability of finding the system in one of the states of $X$ and $\langle B \rangle = \pi_{\overline{X}} = 1 - \pi_X$, leading to the result

$$\left| \frac{d \log \pi_X}{d \log x} \right| \le m(1 - \pi_X). \tag{8}$$

Note the similarity of the right hand side of (8) to the derivative of a Hill function, (6). A key consequence of the support bound is that the effective Hill coefficient $H_{eff}$ is always bounded by the size of the support $m$ (Methods).

**Comparison to the equilibrium case**

To contextualize the support bound, it is valuable to compare to the case of thermodynamic equilibrium, where exact, transparent formulas for sensitivity are often available.

Kinetic schemes describing systems at thermodynamic equilibrium must satisfy the principle of detailed balance[41,42], which is equivalent to the following condition on the rates around any cycle of distinct states ($1 \to 2 \to 3 \to \cdots i \to 1$):

$$\frac{W_{21} W_{32} \cdots W_{1i}}{W_{12} W_{23} \cdots W_{i1}} = 1. \tag{9}$$

For any binding scheme, no matter how complicated, that satisfies detailed balance, the sensitivity to the ligand concentration $x$ is given by a simple expression. If $X$ is a set of states of interest, and $\overline{X}$ is its complement (the set of states not in $X$), then we have (Supplementary Note 1)

$$\frac{d \log \pi_X}{d \log x} = \underbrace{\left[ \langle n_b \rangle_X - \langle n_b \rangle_{\overline{X}} \right]}_{H_{eff}} (1 - \pi_X), \tag{10}$$

where $\langle n_b \rangle_X$ (resp. $\langle n_b \rangle_{\overline{X}}$) is the expected number of ligands bound, conditional on the system being found in one of the states of $X$ (resp. $\overline{X}$).

The right hand side of (10) cannot exceed $n(1 - \pi_x)$, where $n$ is the maximum possible number of ligands that can be bound. Therefore, comparing to the sensitivity of a Hill function (6), we see how this formula (10) refines the observation that, at thermodynamic equilibrium, the effective Hill coefficient cannot exceed the number of binding sites.

The support bound does not require detailed balance, and so it applies to models of nonequilibrium systems. In the form (8), we can compare it directly to (10). We will see that the size of the support $m$ can considerably exceed the number of binding sites $n$, and that this enhanced sensitivity is achievable by nonequilibrium schemes.

### Nonequilibrium MWC and the flagellar motor

Now we turn to bacterial chemotaxis, where the support bound sheds light on the (possibly nonequilibrium) mechanism underlying the sensitive directional switching of the flagellar motor.

In the chemotaxis system of *E. coli*, an array of receptors senses the chemical environment of the cell and controls the intracellular concentration of the phosphorylated protein CheY-P. In turn, the CheY-P concentration controls the direction of rotation of the flagellar motors of *E. coli*−determining whether the bacterium "runs" or "tumbles." The relationship between [CheY-P] and the fraction of the time a motor rotates clockwise, $\pi_{CW}$, is known to be an extremely sensitive one, with studies[43–46] over time reporting increasingly large Hill coefficients, as experimental techniques have more fully isolated a single motor's "input-output relation". Recent measurements, due to Yuan and Berg[47], were fit well to a Hill function with $H \approx 21$.

The underlying mechanism generating this sensitivity is unknown, but is thought to involve the binding of CheY-P to some of the ~ 34 FliM protein subunits of the motor, promoting clockwise rotation. There have been several equilibrium models of cooperative binding proposed for this, including, notably, the Ising-like conformational spread model[48].

But for any equilibrium model, including the conformational spread model, (10) predicts that the Hill coefficient for directional switching is given by the difference in the mean number of bound CheY-P molecules in the clockwise (CW) and counterclockwise (CCW) rotation states. Fukuoka et al.[49] measured a quantity very much like this−finding an average of 13 CheY-P molecules are bound when the motor rotates CW, compared to an average of only 2 bound during CCW rotation. This measurement may be mixed up with intrinsic fluctuations of [CheY-P], but even allowing for this, the difference of these numbers ~11 should still exceed the Hill coefficient (Supplementary Note 2), contradicting the finding $H \approx 21$ of Yuan and Berg. A nonequilibrium mechanism is needed to reconcile these observations.

Other lines of evidence, including observations of the statistics of the time spent in the CW or CCW states between switching events, also point to a nonequilibrium mechanism[20,50]. Tu[19] proposed a simple nonequilibrium model for directional switching. The model, illustrated in Fig. 1c, is a kinetic scheme with the structure of an MWC model−coupling the binding of $n$ ligands (CheY-P) to a global (i.e., concerted) transition between the two motor states (CW or CCW)−except that detailed balance is broken.

Tu assumed a particular form for the rate constants in the model, but here we relax the choice of rate constants, and ask what sensitivity is possible in models with this general, "nonequilibrium MWC" structure. By counting the green states in Fig. 1c, we see that $m = 2n$ for models in this class. Therefore, the support bound (taking $X = $ CW in (8)) constrains the sensitivity of the clockwise bias $\pi_{CW}$ to changes in the CheY-P concentration, $x = $ [CheY-P], as

$$\frac{d \log \pi_{CW}}{d \log[\text{CheY-P}]} \leq 2n(1 - \pi_{CW}), \qquad (11)$$

or $H_{eff} \leq 2n$. This bound can be approached arbitrarily closely in an appropriate limit of transition rates (Supplementary Note 3). In fact, $\pi_{CW}$ can be seen to approach a Hill function with $H = 2n$, $\pi_{CW}(x) \rightarrow x^{2n}/(K^{2n} + x^{2n})$, saturating (11).

In models of this form, $n$ is the difference between the largest and smallest possible number of bound ligands. For a model of the flagellar motor, the simplest interpretation is that $n \approx 34$, the number of FliM subunits. However, Fukuoka et al. found that very high FliM occupancies were rare. If it were the case that the number of bound CheY-P molecules were constrained to never leave the range 2–13, then we could take $n = 13 - 2 = 11$. $2n = 22$ would then be suggestively close to the Hill coefficient measured by Yuan and Berg[47]. However, we cannot exclude the possibility that transient passage through rare states could have an outsized effect on sensitivity. What (11) says is that to explain a Hill coefficient of 21 using a model of this form, it is necessary to allow for a range of least $n = 11$ in the number of ligands bound.

### Proofreading and sensing

Next, we consider the application of the support bound to kinetic proofreading (KP)[4,11] and to the problem of sensing chemical concentrations. These seemingly distinct scenarios are very similar in mathematical structure[22]. The support bound refines known limits for KP schemes[6] and yields constraints on the accuracy of chemical sensing by a nonequilibrium receptor[21].

To frame our discussion of KP, consider an enzyme $E$ that can bind either of two very similar substrates present at equal concentrations−a "right" one $R$ and a "wrong" one $W$. Suppose the enzyme-substrate complex $EW$ has a free energy larger than that of $ER$ by a small amount $\Delta k_B T$. Then, at thermodynamic equilibrium, the so-called error fraction $\eta \equiv [EW]/[ER]$ equals the Boltzmann factor $\eta = b \equiv \exp(-\Delta)$.

KP is a nonequilibrium kinetic scheme, illustrated in Fig. 1d, that can enrich $ER$ over $EW$, amplifying the effect of the small energy difference $\Delta$, while otherwise treating $R$ and $W$ exactly the same. The degree of this amplification can be quantified by a sensitivity−the "discriminatory index" $\nu$, introduced by Murugan[6]:

$$\nu = -\frac{d \log([EW]/[ER])}{d\Delta} = -\frac{d \log([EW]/[E])}{d\Delta}, \qquad (12)$$

which is of a form we can constrain with the support bound (7), taking $x = \exp(\Delta)$. At equilibrium, we must have $\nu = 1$. For the scheme in Fig. 1d, the size of the support is the number of bound states of $W$ that can dissociate, so $m = 2$, and we get $\nu \leq 2$. This upper limit $\nu \rightarrow 2$ can be approached for an appropriate choice of transition rates, corresponding to Hopfield's result that the error fraction can approach $b^2$.

The support bound also recovers the limits of discrimination in more general KP schemes, as studied by Murugan[5,6]. The class of generalizations we consider is illustrated in Fig. 2a. We consider any scheme whose states can be divided into two sets $X$ and $Y$, such that the transitions depending on the parameter $x$ are exactly those crossing $Y$ to $X$.

The KP interpretation of Fig. 2a is that somewhere in $X$ there is a state that represents the unbound enzyme $E$, and somewhere in $Y$ there is a state that represents a bound state $EW$. Somewhere along the paths between these states, there are transitions that cross, in Murugan's[6] language, a "discriminatory fence" and depend on $x = \exp(\Delta)$. Murugan focused on the number $c$ of transitions crossing this "fence," and found $\nu \leq c$. The support bound (7) tells us instead to count the number $m$ of "boundary states" from which these crossing transitions emanate from, yielding $\nu \leq m \leq c$. Saturation of this bound is possible by the "ladder"-like multi-step proofreading schemes discussed in ref. [5].

Harvey et al.[21] sought to understand universal constraints on sensing chemical concentration by studying a model of a general nonequilibrium receptor, illustrated in Fig. 2b. This has exactly the

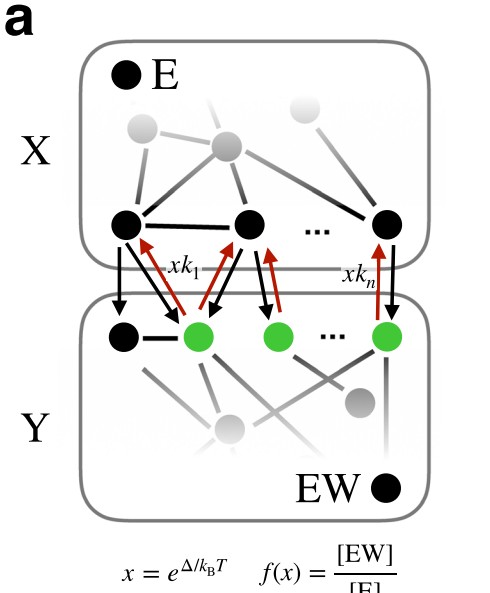
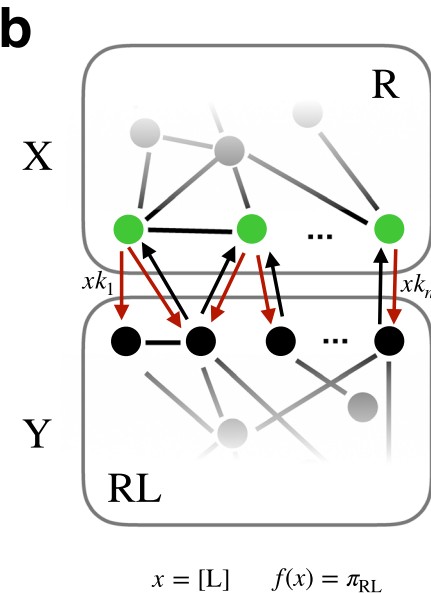

$$x = e^{\Delta/k_B T} \qquad f(x) = \frac{[EW]}{[E]}$$

$$x = [L] \qquad f(x) = \pi_{RL}$$

**Fig. 2 | Common structure of proofreading and sensing models. a** Murugan's generalized proofreading scheme[6], where the key assumption is that there is a "discriminatory fence" dividing the states into two halves, and every transition depending on the energy difference Δ crosses this fence. We are showing just half of a symmetric kinetic scheme—restricting attention to the part of the graph on

which ν depends, which is just the reactions involving bound states containing $W$ (neglecting the totally analogous ones involving $R$). **b** The general receptor model used in ref. [21] to study the sensing of a ligand concentration has the same structure, with the separation between the bound and unbound states of the receptor playing the role of the "fence".

same structure as Murugan's general KP schemes. $X$ now corresponds to states in which the receptor is not bound to a ligand (nonsignaling states), and $Y$ corresponds to those when it is (signaling state), and $x = c$, the concentration of the ligand being sensed.

The key question in sensing is how well $c$ can be estimated. The accuracy is related to a sensitivity that we can constrain using the support bound. To see how this can work, suppose a cell's surface is covered by a number $R_T$ of identical, non-interacting receptors each modeled by (the same) scheme in the form of Fig. 2b, and that they relax to a steady-state distribution over their states subject to fixed external ligand concentration $x = c$. Now consider the number $r$ of the receptors that are in a "signaling state" (one of the states in the set $Y$), at one particular instant. For the sake of example, this will be our "readout"[51] from which we wish to construct an estimate $\hat{c}$ of $c$. The mean of $r$ is a function of $c$ given by $f(c) = R_T \pi_Y(c)$, and if it is invertible, we could take $\hat{c} = f^{-1}(r)$. The error in an estimate so constructed can, under certain assumptions, be approximated as

$$\epsilon_{\hat{c}}^2 \equiv \frac{\text{Var}\,\hat{c}}{c^2} \approx \frac{\text{Var}\,r}{R_T^2 \left(c\frac{d\pi_Y}{dc}\right)^2}. \tag{13}$$

Now, $r$ is a binomial random variable, so $\text{Var}\,r = R_T \pi_Y(1 - \pi_Y)$, and the derivative in the denominator we can bound in terms of the size of the support, $m$,

$$c\frac{d\pi_Y}{dc} = \frac{d\pi_Y}{d\log c} \leq m\pi_Y(1 - \pi_Y) \tag{14}$$

leading to a lower bound on the sensing error in terms of the support:

$$\epsilon_{\hat{c}}^2 \geq \frac{1}{R_T m^2 \pi_X(1 - \pi_X)} \geq \frac{4}{R_T m^2}. \tag{15}$$

The larger the support, the higher is the achievable sensitivity and the lower is the achievable sensing error.

## Unordered binding and nested hysteresis

Finally, we turn to the application of the support bound to models in which identical ligands bind, in any order, to $n$ distinguishable binding sites. In this case, we will see that the support bound gives a limit on sensitivity that is exponential in $n$. We find that this remarkable sensitivity can in fact be achieved, by a nonequilibrium mechanism we call nested hysteresis.

The motivating example in this section will be the regulation of a gene by the binding of copies of a transcription factor (TF) to multiple sites along a DNA molecule (Fig. 3a). Gene expression can be strikingly sensitive to TF concentration. For example, in the *Drosophila* embryo, an exponentially decaying spatial gradient of the TF called Bicoid is transformed into a sharply sigmoidal pattern of Hunchback gene expression across the embryo[52–54]. These observed patterns can be fit to a Hill function with $H \sim 5 - 7$[7,8,23,24]. Many authors have proposed to explain this as a consequence of equilibrium cooperative binding to $5 - 7$ Bicoid binding sites[24,55–57], but this picture is at least clouded by recent theory and experiments which found effects of binding site deletions that were contrary to equilibrium expectations[7,23]. And indeed, especially in eukaryotes, there are many avenues by which energy may be expended in gene regulation, breaking detailed balance[14].

Inspired by this example, Estrada et al.[7] asked what relationships between TF concentration and gene expression could arise from the binding of TFs to distinguishable sites, in any order, without assuming detailed balance (Fig. 3a). In unordered binding, each of $n$ binding sites can be occupied or not, independently of the others, so there are $2^n$ possible states in the kinetic scheme. The allowed transitions are those involving the binding or unbinding of single TF molecule, resulting in a hypercube graph of states and transitions (illustrated in Fig. 3b for $n = 3$). The binding transitions are assumed to have rates linear in the TF concentration, $x$, and all other transition rates are independent of $x$.

What does the support bound say about sensitivity in such models? The support of the perturbation of $x$ in the case of

**a**

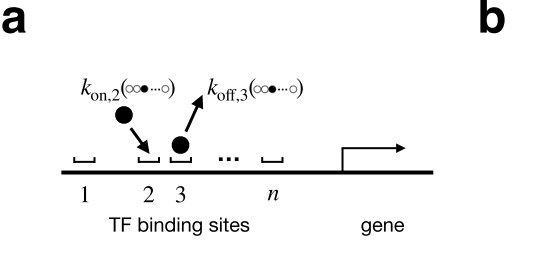

**b**

$n = 3$
$\#(\bullet) = 7$

**Fig. 3 | Unordered binding. a** A gene may be regulated by the binding of transcription factors (TFs) to some number $n$ of sites. The most general case is that of unordered binding to distinguishable sites. The TFs might bind in any order, and

the rates $k_{on}$ and $k_{off}$ of binding and unbinding may be different for each site and depend on the occupancy state of all the other sites. **b** The graph of states and transitions for unordered binding of $n = 3$ copies of a ligand.

unordered binding consists of every single state of the system, except for the fully bound state, in which every binding site is occupied and no more binding can occur. Therefore, the size of the support is $m = 2^n - 1$ and the support bound implies, for example,

$$\frac{d \log \pi_{all}}{d \log x} \le (2^n - 1)(1 - \pi_{all}), \qquad (16)$$

where $\pi_{all}$ is the steady-state probability of the fully bound state—which, following ref. [7], could be identified with the level of gene expression in a picture where transcriptional activation requires binding at all $n$ sites. As discussed earlier, we may also say, more roughly, that $H_{eff} \le 2^n - 1$. Can such exponential-in-$n$ sensitivity actually be achieved? We find that it can be, by a simple mechanism—nested hysteresis—that we describe in the next paragraph. And in fact, by a small further elaboration of the mechanism, it appears that $\pi_{all}$ (viewed as a function of the ligand concentration $x$), can be made as close as desired to a Hill function with $H = 2^n - 1$.

There are two key ingredients in nested hysteresis. Suppose the binding sites are numbered: $1, \ldots, n$. The first ingredient is a hierarchy of timescales, such that binding and unbinding to each successive (higher-numbered) site is much slower than to the (lower-numbered) one before. The second ingredient is a simple rule restricting when binding and unbinding can occur—binding to a site can only happen when all the lower-numbered ones are bound, and unbinding from a site can only happen when all lower-numbered ones are unbound. When binding or unbinding at a site can occur, they occur at some rate and we suppose the ratio of these rates equals $x$ (as if working in units where the dissociation constant equals one).

These rules gives rise to a nested structure, where the dynamics at each binding site depend only on lower-numbered ones. The iterative construction of a kinetic scheme realizing this mechanism for any $n$—incorporating an explicit scale factor $s$ giving rise to the required timescale separation in the limit $s \to \infty$—is illustrated in Fig. 4a. Typical stochastic dynamics of this scheme for $n = 3$ and a finite value of $s$ are shown in Fig. 4b, illustrating its hallmarks—the hierarchy of timescales, and the dependence of the dynamics of higher-numbered sites on the occupancy of the lower-numbered ones.

In the limit of strong timescale separation, we can analytically find the steady-state distribution of nested hysteresis. We give here an intuitive argument, and provide more careful arguments in Supplementary Note 4. To begin, we start by considering the first binding site. This site is independent of all the others, and the ratio of the binding rate to the unbinding rate is $x$, so the steady-state probability that the

first site is bound is

$$\pi(\text{site 1 is bound}) = \frac{x}{1 + x}. \qquad (17)$$

By assumption, binding to the second site can only happen when the first is bound. And, since (again, by assumption) there is a strong timescale separation between these two sites, this amounts to an effective rate of binding to the second site of $kx \times \pi(\text{site 1 is bound}) = \frac{kx^2}{1+x}$, where $k$ is a constant that will drop out. Unbinding happens at an effective rate of $k \times \pi(\text{site 1 is bound}) = \frac{k}{1+x}$. From this it follows that

$$\pi(\text{site 2 is bound}) = \frac{x^2}{1 + x^2}. \qquad (18)$$

Importantly, it is also a consequence of the timescale separation that the sites behave as though they are independent at steady state, in the sense that the probability both are bound is the product of the probabilities that each one is, so that

$$\pi(\text{sites 1 and 2 are bound}) = \left(\frac{x}{1+x}\right)\left(\frac{x^2}{1+x^2}\right). \qquad (19)$$

Now, since binding to the third site can only happen when the first two are bound, we can iterate this argument, leading to (i.e., by induction),

$$\pi(\text{site } i \text{ is bound}) = \frac{x^{2^{i-1}}}{1 + x^{2^{i-1}}}, \qquad (20)$$

for all $i = 1, \ldots, n$, from which we can find the steady-state probability of any occupation state of the sites, including the fully bound state, which has probability

$$\pi_{all} = \pi(\text{all sites bound}) = \prod_{i=1}^{n} \frac{x^{2^{i-1}}}{1 + x^{2^{i-1}}} = \frac{x^{2^n - 1}}{\sum_{j=0}^{2^n - 1} x^j}. \qquad (21)$$

This expression saturates (16) only when $\pi_{all}$ is very small, and it is not a Hill function. However, we can get one if we now stabilize the two extreme occupation states—the fully bound and totally empty states—by slowing the rates of all transitions leaving them. We can accomplish this by scaling the exit rates from these states by a

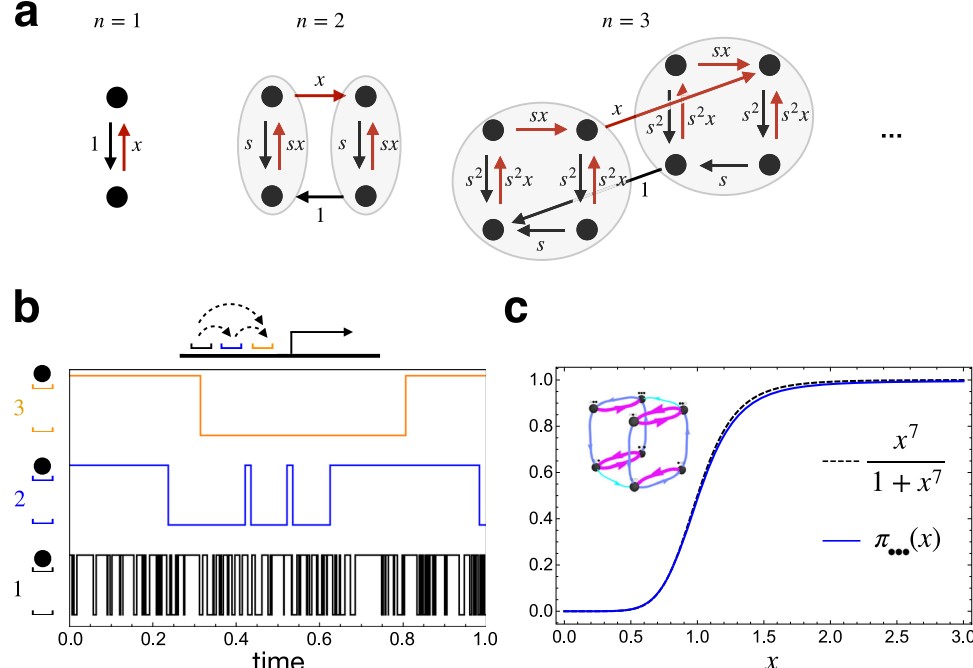

**Fig. 4 | Nested hysteresis. a** Iterative construction of the kinetic scheme of nested hysteresis, generalizable to any value of $n$. In each diagram, the gray ovals indicate the subsystems (corresponding to binding and unbinding to the first $n-1$ sites), which are assumed to relax much faster than the other transitions. **b** Occupation over time for each of $n = 3$ binding sites (black, blue, and orange), in a particular

stochastic realization of nested hysteresis (parameters: $n = 3$, $s = 10$, $x = 2$). Dotted arrows in the schematic indicate the pattern of influences between the binding sites. **c** Illustration of a parameter choice (inset, color indicates rate—magenta = $10^4$, blue = 100, cyan = 1) for which the dependence on $x$ of the probability of the fully bound state approaches a Hill function with $H = 7$ very closely.

factor of $q$, leading to

$$\pi_{\text{all}} = \frac{x^{2^n - 1}}{1 + q\left(\sum_{j=1}^{2^n - 2} x^j\right) + x^{2^n - 1}} \tag{22}$$

which approaches a Hill function with $H = 2^n - 1$ as $q \to 0$, and saturates support bound (16) for all values of $\pi_{\text{all}}$ simultaneously. Further details can be found in Supplementary Note 4.

The possibility of exponential-in-$n$ sensitivity—the optimal nonequilibrium sensitivity achievable with unordered binding to $n$ sites—was missed in the prior numerical work[7,8] which instead seemed to suggest that, e.g., for $n = 3$, $H_{\text{eff}} \leq 5$. An explicit parametric choice, for the case $n = 3$, approaching very close to the Hill function with $H = 2^3 - 1 = 7$ is illustrated in Fig. 4c.

Do living things actually use a mechanism like nested hysteresis? In our view, this is a very exciting open question raised by our work. One place to look could be in the regulation of distant genes by TF binding to enhancer sequences. Our understanding of mechanism here remains unsettled. Equilibrium[58] and nonequilibrium[17,18] models of enhancer action have been proposed recently, and the character and role of enhancer-promoter loops remains under active experimental study and debate[59–61].

As argued by Grah et al.[17], focusing attention on mechanisms that are optimal in some sense is one way to tackle the huge space of candidate models for enhancers. Nested hysteresis is an example of a "sensitivity-optimal" mechanism. And we note that at a very coarse level, the necessary ingredients for nested hysteresis are present: coordination of multiple enhancers at different genomic distances from a promoter could supply the required nested hierarchy of timescales (e.g., ideal polymer looping times often scale like the length squared[62,63]), and ATP-dependent loop extrusion[64] or chromatin remodeling could provide the requisite nonequilibrium driving.

## Discussion

The idea that structure determines function suffuses biology. In the molecular realm, if the conditions of thermodynamic equilibrium prevail, an important aspect of function, sensitivity, is tightly constrained by the most basic structural property—system size. This general physical fact is most familiar in biophysics as the statement that the Hill coefficient for equilibrium binding of a ligand cannot exceed the number of binding sites.

But living things are not at thermodynamic equilibrium, and today at the frontier of molecular biology we are increasingly led to consider this in our models[14,17]. In this work, we have shown that a structural feature of our models—the size of the support of a perturbation—always limits the sensitivity, at or away from equilibrium. Considering several contexts in which sensitivity is important, we show that our findings unify and extend our understanding of diverse biophysical examples. Table 1 summarizes these results.

Importantly, for any scheme the size of the support of any perturbation is less than the number of system states. This means that the effective Hill coefficient is always less than the number of system states. In the cases considered above, the bound in terms of the support is saturatable or "tight"—it can be approached as closely as desired with an appropriate choice of transition rates.

The possibility of exponential-in-$n$ sensitivity, revealed by nested hysteresis, has implications for our understanding of biomolecular "condensates"[65–68], aggregations of macromolecules increasingly implicated in key biological processes. Some of the proposed functions of these condensates, such as enhancing specificity of reactions or signaling, amount to increasing the sensitivity to a small change in chemical composition or concentration[69]. This idea is well-grounded in the familiar equilibrium logic (e.g., reflected in results like (10))—achieving a switch-like response with, say $H_{\text{eff}} \approx 100$, requires the cooperation of hundreds of molecules, as might be seen in a condensate.

**Table 1 | Support bound compared to the bounds on sensitivity that hold at thermodynamic equilibrium (Eq.), for three classes of models we have discussed**

| Model class | Eq. bound | Support bound | Saturatable? |
|---|---|---|---|
| KP-like, $n$ boundary states | 1 | $n$ | ✓ |
| MWC-like, $n$ sites | $n$ | $2n$ | ✓ |
| Unordered binding, $n$ sites | $n$ | $2^n - 1$ | ✓ |

But nested hysteresis shows that the story is not quite complete—away-from-equilibrium just e.g., 7 molecules could supply a sensitivity that would require the cooperation of $127 \approx 2^7 - 1$ molecules at equilibrium! The large size of a molecular aggregation cannot be accounted for merely by postulating selection for sensitivity—some other cost or constraint must also be part of the story.

One possibility is that the constraint is time. Perhaps all mechanisms that achieve exponential-in-$n$ sensitivity are just too slow for many uses, whether the response to heat shock[69], or gene regulation in a fast-dividing *Drosophila* embryo[8]. Nested hysteresis is "slow," in the sense that it demands a huge gulf between the fastest and slowest rates in this system. Could there be, for any mechanism, a "number-time tradeoff" for achieving a given sensitivity, where involving fewer particles to achieve a given sensitivity is possible but requires more time? If such a law could be framed with the same degree of generality as the support bound, it would be another powerful tool to help us make sense of the nonequilibrium machinery within living cells.

## Methods

### Logarithmic sensitivity, the Hill coefficient, and fold-change amplification

Here we review the relationships between the logarithmic sensitivity and other measures of sensitivity that might be reported or measured in an experiment, especially "the Hill coefficient".

If $f(x)$ were a Hill function (1), then

$$\frac{d \log f(x)}{d \log x} = H\left(\frac{K^H}{K^H + x^H}\right) = H(1 - f(x)). \tag{23}$$

We note two simple facts about this expression. First, the logarithmic sensitivity achieves its maximum value, $H$, when $x$ (and so $f(x)$) is very small. Second, at the midway point $x = K$, where $df(x)/d \log x$ reaches its maximal value $H/4$, the logarithmic sensitivity is $H/2$.

As mentioned in the main text, in general, functions of interest will not actually be Hill functions, but it is common nevertheless to report a Hill coefficient or an "effective Hill coefficient," $H_{\text{eff}}$. There are several different quantities—we will describe several below—sometimes called the effective Hill coefficient, and which give $H_{\text{eff}} = H$ in the case of the Hill function, but which in general are not equivalent. We will later see that the size of the support bounds them all.

First, suppose $f(x)$ is not a Hill function, but that it is known exactly, or at least, we can find its derivative. Then one approach is to define $H_{\text{eff}}$ directly as the logarithmic sensitivity at some point, in analogy to how $H$ controls the sensitivity of the Hill function. For example,

$$H_{\text{eff}} = 2 \frac{d \log f(x)}{d \log x}\bigg|_{x = x^*} \tag{24}$$

where $x^*$ is the value of $x$ at which $f(x)$ is halfway between the smallest and largest value it can assume. This definition has been used to quantify the sensitivity of non-Hill sigmoidal functions arising from theoretical models (e.g., refs. [8,19,30]).

In an experimental context, it is very common to fit a Hill function to data (e.g., averaged observations) that are purported to reflect a functional relationship $f(x)$, and to report the fit parameter $\hat{H}$ as the Hill coefficient. Often this is informative, but as a matter of principle, two functions can have radically different derivatives even if the function values are very close everywhere (e.g., if one function exhibits very high frequency but low amplitude oscillations). This means that, even if the fit is very good, relations based on analogy to (6), such as that $\hat{H}/2 = d \log f(x)/d \log x$ at the midpoint, can fail dramatically.

A different measure of sensitivity—the amplification of a fold-change in the input—provides a solution to this problem. Suppose that for some value $x_0$ of the input parameter $x$, scaling by a factor $a$ scales the output by $b$, so $f(ax_0) = bf(x_0)$. Then the quotient $\log(b)/\log(a)$ can be thought of as a discrete approximation of the derivative defining the logarithmic sensitivity. And if $f(x)$ is differentiable everywhere, then by the mean value theorem, there must be a value $x^*$ of $x$ for which

$$\frac{d \log f(x)}{d \log x}\bigg|_{x = x^*} = \frac{\log b}{\log a}. \tag{25}$$

This means that careful measurement of any two points on the input-output curve ($x$ versus $f(x)$) witnesses the (local, infinitesimal) logarithmic sensitivity somewhere. Importantly—unlike in the case of fitting to a Hill function—if error in the measurements is very low, then they are also telling us the derivative for some value of $x$ very accurately.

Equation (25) leads us to another common definition of the effective Hill coefficient[29]:

$$H_{\text{eff}} = \frac{\log 81}{\log(S_{0.9}/S_{0.1})}, \tag{26}$$

where $S_{0.9}$ and $S_{0.1}$ are the values of the input variable (in our case, $x$) required to get 90% and 10% (respectively) of the maximum value of the output variable (in our case, $f(x)$). Note that (26) is like (25) with $a = S_{0.9}/S_{0.1}$ and $b = 9$. It implies that somewhere between $S_{0.1}$ and $S_{0.9}$ there is a logarithmic sensitivity of $H_{\text{eff}}/2$.

There is yet another common definition, specific to models of binding. Suppose $x$ is the concentration of a ligand and $\langle n_b \rangle(x)$ is the expected number of sites bound by a ligand out of a total of $n$ possible binding sites. It is common then, to take

$$H_{\text{eff}} = \frac{d}{d \log x} \log\left(\frac{\langle n_b \rangle}{n - \langle n_b \rangle}\right), \tag{27}$$

or to report, as the Hill coefficient, the slope of a line fitted to $x$ versus $(\langle n_b \rangle)/(n - \langle n_b \rangle)$ data on a log-log plot.

As mentioned above, all these definitions of $H_{\text{eff}}$ are inequivalent in general. For example, for totally noncooperative binding to $n = 2$ binding sites, we have $\langle n_b \rangle = 1 \times 2x/(1+x)^2 + 2 \times x^2/(1+x)^2 = 2(x/(1+x))$. In this case, (27) gives 1, as does (26), if we take $f(x) = \langle n_b \rangle$. However, taking instead $f(x) = (x/(1+x))^2$—the fraction of the time both sites are occupied—we get $H_{\text{eff}} \approx 1.17$ from (24) and $H_{\text{eff}} \approx 1.19$ from (26).

### Proof of the support bound

Here we give a proof of the support bound, (7). The technical tool we rely on is the Markov chain tree theorem (MTT, also called "matrix-tree theorem," see refs. [31–36] for details), which gives an explicit algebraic expression for the steady-state distribution $\pi$ of a kinetic scheme in terms of the spanning trees of the associated graph $G$:

$$\pi_k = \frac{1}{Z} \sum_{\substack{\text{spanning trees of } G \\ \text{oriented to } k}} \prod_{\text{tree edges } i \to j} W_{ji}, \tag{28}$$

where $Z$ is the normalization constant, and a spanning tree of $G$ is a connected subgraph of $G$ that includes every vertex but has no cycles. In words, the right hand side gives a recipe to find the steady-state probability of a state $k$. It says to consider each spanning tree of $G$ and orient all its edges (choose their direction) so they point towards $k$, which is called the root of the tree. Then, for each such oriented tree, multiply together the transition rates associated to all its directed edges. Then, add up the products so formed. The result is proportional to $\pi_k$, up to overall normalization.

The sum over spanning trees looks forbidding, but to prove the support bound we rely on only two facts, which follow from (28) simply. The first fact is that every term in that sum is a positive monomial, being a product of nonzero transition rates. This means the ratio of any positive observables is a ratio of polynomials in $x$:

$$\frac{\langle A \rangle_\pi}{\langle B \rangle_\pi} = \frac{\sum_{i=a_{min}}^{a_{max}} k_{i-a_{min}} x^i}{\sum_{j=b_{min}}^{b_{max}} q_{j-b_{min}} x^j} = x^{a_{min}-b_{min}} \frac{\sum_{i=0}^{a_{max}-a_{min}} k_i x^i}{\sum_{j=0}^{b_{max}-b_{min}} q_j x^j} \quad (29)$$

where $k_i$ and $q_j$ are positive quantities that do not depend on $x$, and $a_{max}$, $a_{min}$, $b_{max}$, and $b_{min}$ are nonnegative integers.

Differentiating this expression we find

$$\frac{d \log \langle A \rangle_\pi / \langle B \rangle_\pi}{d \log x} = (a_{min} - b_{min})$$
$$+ \left( \frac{\sum_{i=0}^{a_{max}-a_{min}} i k_i x^i}{\sum_{i=0}^{a_{max}-a_{min}} k_i x^i} - \frac{\sum_{j=0}^{b_{max}-b_{min}} j q_j x^j}{\sum_{j=0}^{b_{max}-b_{min}} q_i x^j} \right). \quad (30)$$

The second term in brackets is not less than $-b_{max} + b_{min}$ and not more than $a_{max} - a_{min}$, so we get

$$a_{min} - b_{max} \le \frac{d \log \langle A \rangle_\pi / \langle B \rangle_\pi}{d \log x} \le a_{max} - b_{min}. \quad (31)$$

The second fact about (28) that we need is that each oriented spanning tree has at most one directed edge emanating from each vertex (it has none coming of its root). Recalling the definition of support, it follows that each monomial in (28) picks up at most $m$ factors of $x$, where $m$ is the size of the support of the perturbation. This means $a_{max}$ and $b_{max}$ are both no greater than $m$, which leads to

$$\left| \frac{d \log \langle A \rangle_\pi / \langle B \rangle_\pi}{d \log x} \right| \le m. \quad (32)$$

### Proof that the effective Hill coefficient is bounded by size of support

Here we show that the size of the support bounds the effective Hill coefficient, for all three definitions given in the first section of the Methods. In two cases, this is very easy–to see it for the "binding" definition (27) of $H_{eff}$, we simply apply (7) with $A = n_b$ and $B = n - n_b$. For the definition (24), applying the corollary (8) is sufficient. It is hardest to see that the size of the support bounds the effective Hill coefficient when the latter is defined according to "nonlocal" definition (26) (in the main text, reproduced here),

$$H_{eff} = \frac{\log(81)}{\log(S_{0.9}/S_{0.1})}. \quad (33)$$

Suppose we are interested in the sensitivity properties of a function $f(x)$ which is positive, monotonically increasing in the parameter $x$, and bounded above by a value $f_{max}$. These assumptions are effectively required to be able to apply (26), e.g., because the definition presupposes the existence and uniqueness of the values $S_{0.1}$ and $S_{0.9}$,

which, recall, are the values of the input variable $x$ for which $f(x)$ achieves 10% and 90%, respectively, of its maximum range. We additionally suppose that $f(x)$ is a positive observable of some kinetic scheme, which implies that $f_{max} - f(x)$ is a positive observable as well.

The support bound then gives

$$\frac{d \log \left( \frac{f(x)}{f_{max} - f(x)} \right)}{d \log x} \le m, \quad (34)$$

where $m$ is the size of the support of the perturbation of $x$. Note that since $f(x)$ is increasing the left hand side is positive. Then, define $z = \log x$ (we assume, as we do throughout this work, that $x > 0$), and write $z_{0.1} = \log S_{0.1}$ and $z_{0.9} = \log S_{0.9}$. Now we integrate the inequality

$$\int_{z_{0.1}}^{z_{0.9}} \frac{d \log \left( \frac{f(x)}{f_{max} - f(x)} \right)}{dz} \, dz \le \int_{z_{0.1}}^{z_{0.9}} m \, dz, \quad (35)$$

which yields

$$\log \left( \frac{0.9 f_{max}}{f_{max} - 0.9 f_{max}} \right) - \log \left( \frac{0.1 f_{max}}{f_{max} - 0.1 f_{max}} \right) \le m(z_{0.9} - z_{0.1}) \quad (36)$$

$$\log(0.9/0.1) - \log(0.1/0.9) \le m(z_{0.9} - z_{0.1}) \quad (37)$$

from which it follows that $\log(81) \le m \log(S_{0.9}/S_{0.1})$, which implies $H_{eff} \le m$, for the definition (26) of $H_{eff}$, as desired.

## Data availability

No datasets were generated or analyzed in this study.

## Code availability

A *Mathematica* notebook[70] accompanying our discussion of nested hysteresis in Supplementary Note 4 is available at https://github.com/jaowen/nested-hysteresis.

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

## Acknowledgements

J.A.O. thanks Leonid Mirny for advice and support. J.M.H. acknowledges support from the National Science Foundation under Grant No. 2142466.

## Author contributions

J.A.O. and J.M.H. conceived the research, carried out the research, and wrote the manuscript together.

## Competing interests

The authors declare no competing interests.
