## [Peer Review File · Nature Communications]

Size limits the sensitivity of kinetic schemesREVIEWERS' COMMENTS

Reviewer #1 (Remarks to the Author):

In this paper the authors derive a general bound on the sensitivity of a biological function described by a kinetic model to changes in a parameter, like the concentration of a molecule. Their bound extends the well known one for models that assume thermodynamics equilibrium to the general context of out of equilibrium models. A particularly intriguing theoretical result is their demonstration of an out of equilibrium model that leads to a Hill coefficient (which they use as the canonical measure of sensitivity of response) that is exponential in the number of binding sites, something that previous numerical studies have not identified. In addition, the paper investigates, in the context of their mathematical result, specific biological models pertaining to switching of the bacterial flagellar motor, proofreading and transcription factor binding to DNA sites. In all cases they describe interesting new results or new interpretations of experimental results (like the sensitivity of the switching of the flagellar motor to changes in concentration of CheY-P), which I found quite interesting and illuminating. Overall, the paper is very well written, the results are clearly described and they are quite interesting, so I am very enthusiastically in support of publication.

I had only two minor suggestions to the authors:

1. The exponential dependence of the Hill coefficient to the number of binding sites is an interesting result. While they give a nice example of how this might come about I was curious if they might further speculate about the biological context in which something like this might be used.
2. I think the title could be a little more informative without sacrificing brevity too much. In particular, I was not sure when seeing the title what "Size" refers to. Might something like "Network size" be more descriptive?

Jane Kondev

Brandeis University

Reviewer #2 (Remarks to the Author):

In this paper the authors derive a very interesting bound for the "sensitivity" of various non-equilibrium processes. The derivation relies on a graph theoretic description of non-equilibrium steady states and it generalizes equilibrium notions of sensitivity going back to the Hill coefficient. The authors demonstrate their results on a variety of systems. For instance, their result suggests how experimental data from E Coli Flagellar experiments measuring a Hill like coefficient actually constrains the nature of the underlying Markov state representation. This is a very satisfying result as it is typically tough to obtain such constraints without resorting to effective equilibrium like approximations. The authors bound allows them to solve this long standing problem in a very creative way ! I can recommend this work for publication without any hesitation.

My only comment is related to the nested hysteresis idea. It seems like the authors are putting forward a proposal but it is as yet not as well supported by experimental data. It will be great if the authors can perhaps comment more on this so that the peices that are speculative are clear to the reader.

REVIEWERS' COMMENTS

Our replies in blue.

We are extremely grateful to both reviewers for their time and comments!

Reviewer #1 (Remarks to the Author):

In this paper the authors derive a general bound on the sensitivity of a biological function described by a kinetic model to changes in a parameter, like the concentration of a molecule. Their bound extends the well known one for models that assume thermodynamics equilibrium to the general context of out of equilibrium models. A particularly intriguing theoretical result is their demonstration of an out of equilibrium model that leads to a Hill coefficient (which they use as the canonical measure of sensitivity of response) that is exponential in the number of binding sites, something that previous numerical studies have not identified. In addition, the paper investigates, in the context of their mathematical result, specific biological models pertaining to switching of the bacterial flagellar motor, proofreading and transcription factor binding to DNA sites. In all cases they describe interesting new results or new interpretations of experimental results (like the sensitivity of the switching of the flagellar motor to changes in concentration of CheY-P), which I found quite interesting and illuminating. Overall, the paper is very well written, the results are clearly described and they are quite interesting, so I am very enthusiastically in support of publication.

I had only two minor suggestions to the authors:

1. The exponential dependence of the Hill coefficient to the number of binding sites is an interesting result. While they give a nice example of how this might come about I was curious if they might further speculate about the biological context in which something like this might be used.

We have added a brief discussion of this immediately before the Discussion section—using the example of enhancer-stimulated gene expression. This is an area where understanding of mechanism, and the global logic behind regulation by enhancers, remains seriously unsettled. There is ample scope for involvement of nonequilibrium driving, and a natural hierarchy of timescales—two important requirements for nested hysteresis, suggesting it could at least be interesting to reassess facts about enhancer action with in mind the possibility that nested hysteresis could be at work.

2. I think the title could be a little more informative without sacrificing brevity too much. In particular, I was not sure when seeing the title what “Size” refers to. Might something like “Network size” be more descriptive?

We appreciate this point and were quite conflicted about it, but would prefer to stick with simply “Size”.

The number of states, or the size of the kinetic scheme, does limit sensitivity, as we note in the Introduction and Discussion. And this is a straightforward and correct interpretation of our title. However, a big part of the paper is that we are also introducing a finer bound in terms of a different kind of “size”—the size of the support of a perturbation. It is still very much a “size” in the sense that it is a count of states, but it can deviate considerably from the network size—e.g. for a receptor with N conformations that can bind one ligand, the network size might be $2N$, but the bound on the sensitivity to ligand concentration is N .

We would like the title to gesture also towards this more precise notion without being technical, and we feel “Size” strikes the right balance.

Jane Kondev
Brandeis University

Reviewer #2 (Remarks to the Author):

In this paper the authors derive a very interesting bound for the "sensitivity" of various non-equilibrium processes. The derivation relies on a graph theoretic description of non-equilibrium steady states and it generalizes equilibrium notions of sensitivity going back to the Hill coefficient. The authors demonstrate their results on a variety of systems. For instance, their result suggests how experimental data from E Coli Flagellar experiments measuring a Hill like coefficient actually constrains the nature of the underlying Markov state representation. This is a very satisfying result as it is typically tough to obtain such constraints without resorting to effective equilibrium like approximations. The authors bound allows them to solve this long standing problem in a very creative way ! I can recommend this work for publication without any hesitation.

My only comment is related to the nested hysteresis idea. It seems like the authors are putting forward a proposal but it is as yet not as well supported by experimental data. It will be great if the authors can perhaps comment more on this so that the peices that are speculative are clear to the reader.

We have added a brief discussion of this immediately before the Discussion section. We now emphasize that it is an open question whether or not living things use nested hysteresis at all. We note that enhancer-stimulated gene expression involves some of the required elements for nested hysteresis.